# The Association between Carotenoids and Head and Neck Cancer Risk

**DOI:** 10.3390/nu14010088

**Published:** 2021-12-26

**Authors:** Adam Brewczyński, Beata Jabłońska, Marek Kentnowski, Sławomir Mrowiec, Krzysztof Składowski, Tomasz Rutkowski

**Affiliations:** 1I Radiation and Clinical Oncology Department of Maria Skłodowska-Curie National Research Institute of Oncology, Gliwice Branch, 44-102 Gliwice, Poland; Adam.Brewczynski@io.gliwice.pl (A.B.); marek.kentnowski@io.gliwice.pl (M.K.); Krzysztof.Skladowski@io.gliwice.pl (K.S.); Tomasz.Rutkowski@io.gliwice.pl (T.R.); 2Department of Digestive Tract Surgery, Medical University of Silesia, 40-752 Katowice, Poland; mrowasm@poczta.onet.pl

**Keywords:** oral cavity cancer, pharyngeal cancer, oropharyngeal cancer, nasopharyngeal cancer, laryngeal cancer, *α*-carotene, *β*-carotene, *β*-cryptoxanthin, zeaxanthin, lycopene, lutein

## Abstract

Head and neck cancer (HNC) includes oral cavity cancer (OCC), pharyngeal cancer (PC), and laryngeal cancer (LC). It is one of the most frequent cancers in the world. Smoking and alcohol consumption are the typical well-known predictors of HNC. Human papillomavirus (HPV) is an increasing etiological factor for oropharyngeal cancer (OPC). Moreover, food and nutrition play an important role in HNC etiology. According to the World Cancer Research Fund and the American Institute for Cancer Research, an intake of non-starchy vegetables and fruits could decrease HNC risk. The carotenoids included in vegetables and fruits are well-known antioxidants which have anti-mutagenic and immune regulatory functions. Numerous studies have shown the relationship between carotenoid intake and a lower HNC risk, but the role of carotenoids in HNC risk is not well defined. The goal of this review is to present the current literature regarding the relationship between various carotenoids and HNC risk.

## 1. Introduction

Head and neck cancer (HNC) includes oral cavity cancer (OCC), pharyngeal cancer (PC), and laryngeal cancer (LC) [1]. It is the sixth most frequent cancer in the world and this disease is recognized in half a million patients every year [2]. Smoking and alcohol are the typical well-defined predictors of HNC. Human papillomavirus (HPV) is an increasing etiological factor for oropharyngeal cancer (OPC) [1,3,4]. Moreover, diet and nutrition are essential in HNC etiology [5]. According to the World Cancer Research Fund and the American Institute for Cancer Research (2007), the intake of non-starchy vegetables and fruits could decrease HNC risk [1,5]. The carotenoids included in vegetables and fruits are well-defined antioxidants which have anti-mutagenic and immune regulatory functions [1,6]. Some carotenoids, as such antioxidants, decrease the adverse results of reactive oxygen species (ROS). ROS are associated with numerous disorders, including cancer, cardiovascular and neurodegenerative disorders, as well as aging. The important roles of carotenoids are in gene regulation, angiogenesis, and apoptosis, as well as the regulation of immune function, which has been reported in the literature [6]. The general effects of carotenoids are presented in Figure 1. Numerous authors have shown the relationship between carotenoid consumption and a lower HNC risk, but the role of carotenoids in HNC risk is not well defined [1,7,8,9,10,11,12,13,14,15,16,17,18,19,20,21,22,23,24,25,26,27,28,29,30,31,32,33,34,35,36,37,38,39,40,41,42,43,44,45,46].

The goal of this review is to present the current literature regarding the relationship between various carotenoids and HNC risk.

## 2. Characteristics of Carotenoids

### 2.1. General Characteristics

So far, more than seven hundred carotenoids have been described. About fifty carotenoids are found in a typical human diet. Carotenoids, chemically tetraterpenes, are the most common lipid-soluble pigments, which are responsible for yellow, orange, or red colors. According to their chemical composition, they are divided into oxygenated carotenoids such as beta-cryptoxanthin (*β*-cryptoxanthin), lutein, zeaxanthin, astaxanthin, fucoxanthin, canthaxanthin, which are known as xanthophylls, and hydrocarbon carotenoids such as alpha-carotene (*α*-carotene), beta-carotene (*β*-carotene), and lycopene, which are known as carotenes [1,47]. The chemical structures of the most common carotenoids are presented in Figure 2.

These substances are exogenous nutrients for mammals, including humans, because they cannot produce them. Therefore, a dietary consumption of carotenoids is needed. Vegetables and fruits are the main dietary sources of carotenoids. They can be also found in other dietary components, such as herbs, legumes, cereals, algae, egg yolks, mammalian milk and tissues, seafood, colorants, and in the artificial form of supplements [47,48]. The usual dietary consumption of carotenoids ranges from 0 to 10 mg per day [48,49]. Carotenoids are involved in numerous processes, such as photosynthesis, communication between species by color signaling, nutrition, and others [47,48,49,50]. In plants and algae, they absorb light energy for use in photosynthesis, and they provide photoprotection via non-photochemical quenching. In animals, they play an important role in supporting oxygen in its transport, storage, and metabolism. In humans, carotenoids containing unsubstituted beta-ionone rings (including *β*-carotene, *α*-carotene, *β*-cryptoxanthin, and γ-carotene) have vitamin A activity and they can be converted to retinol. In the eye, lutein, *meso*-zeaxanthin, and zeaxanthin are present as macular pigments important for visual function and retina protection. Carotenoids such as antioxidants collaborate with other biomolecules, such as proteins and lipids, to enhance its activity. In plants, they protect the cells from extra UV light that is not useful for photosynthesis, as it induces stress on the plant cells. Apocarotenoids are responsible for plant aromas, colors, and phytohormone production. They play a role in producing signals among the plant cells [47,48,49,50]. In humans, dietary carotenoids can increase bone density and reduce the risk of bone fractures by its positive effect on the bone mineral status. They may help to decrease osteoporosis in patients. Additionally, anti-inflammatory, antibacterial, antiviral, and anticancer effects, as well as preventive effects for cardiovascular and neurodegenerative diseases has been proven in humans. Carotenoids, as precursors of vitamin A, play a role in human eyes as it is component of rhodopsin, which facilitates the transfer of energy from photons of light to electrochemical signals. Oxygenated carotenoids (lutein and zeaxanthin) are found in the macular region of the retina and are responsible for sharp and detailed vision, which also acts as filters for blue light from screens and scavenges the free radicle oxygen from the retina. A decrease in the risk of type 2 diabetes has been also reported. The other carotenoids roles in humans that are postulated in the literature are the following: UV radiation protection, antituberculosis, proliferating agent roles, and regenerative liver roles [51].

Concentrations of circulating carotenoids are useful markers of dietary fruit and vegetable intake. A high fruit and vegetable (FAV) intake correlates with high plasma concentrations of carotenoids [52,53]. According to the world literature, concentrations of circulating carotenoids are different depending on gender: higher concentrations of carotenoids were reported in women compared to men, independently of dietary intake. In the comparative study by Allore et al. [52], concentrations of circulating carotenoids were compared between two groups including 155 males and 110 females. The authors demonstrated that plasma concentrations of total carotenoids were negatively correlated with body weight (r = −0.47, *p* = 0.0001) and waist circumference (r = −0.46, *p* = 0.0001), and were positively correlated with plasma LDL-cholesterol (r = 0.49, *p* = 0.0001) and HDL-cholesterol (r = 0.50, *p* = 0.0001) levels. This study also confirmed the above-mentioned gender difference regarding the plasma concentration of total carotenoids. In women, significantly higher plasma concentrations of total carotenoids were noted, despite a significantly lower dietary carotenoid intake [52].

### 2.2. Common Types of Carotenoids

#### 2.2.1. *β*-Carotene

This is the most common carotenoid present in numerous vegetables and fruits. This tetraterpenoid is a precursor of vitamin A after its consumption and digestion. It is used as an antioxidant as well as a natural colorant in the food industry [46,47,48,49].

#### 2.2.2. Lutein (LUT)

This 3R,3′R,6′R-*β*,ε-carotene-3,3′-diol is a natural carotenoid present in numerous vegetables and fruits, as well as being produced in algae. It is an orange-yellow xanthophyll which is frequently added to food and used as a colorant in the food industry [47,48,49,50,54].

#### 2.2.3. Zeaxanthin (ZX)

This *ββ*-carotene-3,3′-diol is a yellow-orange xanthophyll present commonly in egg yolks and dark green leafy vegetables. Similar to LUT, ZX is accumulated in the central retina, and it provides protection to the retina against damage by intense light [47].

#### 2.2.4. Astaxanthin (ATX)

This 3,3′-dihydroxy-*β*,*β*′-carotene-4,4′-dione is a xanthophyll commonly found in microalgae, marine invertebrates, and some fishes (such as salmon and trout). Moreover, it is found in the feathers of some birds, where it contributes to their red-orange color [47].

#### 2.2.5. Fucoxanthin (FX)

This 3*S*,3′*S*,5*R*,5′*R*,6*S*,6′*R*,8′*R*-3,5′-dihydroxy-8-oxo-6′,7′-didehydro-5,5′,6,6′,7,8-hexahydro-5,6-epoxy-*β*,*β*-caroten-3′-yl acetate is an orange-colored xanthophyll that is commonly present in marine environments, as well as macro- and microalgae [47].

#### 2.2.6. *β*-Cryptoxanthin (BCX)

This (1R)-3,5,5-trimethyl-4-[(1*E*,3*E*,5*E*,7*E*,9*E*,11*E*,13*E*,15*E*,17*E*)-3,7,12,16-tetramethyl-18-(2,6,6-trimethylcyclohexen-1-yl)octadeca-1,3,5,7,9,11,13,15,17-nonaenyl]cyclohex-3-en-1-ol is an orange xanthophyll commonly present in fruits, such as orange rind, papaya, and apples, as well as butter and egg yolks [47].

#### 2.2.7. Canthaxanthin (CX)

This *β*,*β*-carotene-4,4′-dione is a red-orange xanthophyll frequently used as a cosmetic and food colorant and is often added to food [47].

#### 2.2.8. Lycopene

This is an orange-red carotenoid found in several vegetables and fruits. As opposed to *β*-carotene, it is not a precursor of a vitamin A in the human body. It can be found in several stereoisomeric forms. Naturally, lycopene occurs in the structural form of “trans” type isomers, but under the influence of heat sources or light irradiation, it changes its structure into cis isomers (mainly in positions 5, 9, 13, and 15). Tomatoes, as well as tomato-based sauces and juices, are the most abundant sources of this compound for humans. [55,56].

## 3. Methods of the Literature Search

The PubMed database was reviewed. The search terms and mesh headings were as follows: “head and neck cancer”, “oral cavity cancer”, “oropharyngeal cancer”, “laryngeal cancer”, “carotenoids”, “alpha-carotene”, “beta-carotene”, “lycopene”, “beta-cryptoxanthin”, “lutein”, and “zeaxanthin”, as well as “carotenoids”, “cancer”, and “HPV”. Selected articles presenting significant associations between carotenoids and HNC risk are presented and discussed in this review. The observational studies are followed by interventional studies in our review. Systematic reviews and meta-analyses are presented and discussed as first publications, followed by prospective randomized cohort studies and case-control studies. The cohort and case-control studies are divided into two main groups: investigating the relationship between the intake of carotenoids from dietary sources and HNC risk, and reporting the relationships between the intake and serum levels of specific types of carotenoids and HNC risk. Moreover, studies on the relationship between carotenoids and survival in HNC patients are presented. In each of the above subgroups, studies are presented and discussed in chronological order.

## 4. Literature Review

### 4.1. Studies Presenting the Relationship between Carotenoids and HNC Risk

#### 4.1.1. Systematic Reviews and Meta-Analyses

Leoncini et al. published two meta-analyses regarding the relationship between all carotenoids and HNC [1,2]. The meta-analysis of 16 epidemiological studies regarding the dietary consumption of specific carotenoids [1] revealed a lower HNC risk associated with the consumption of *β*-carotene equivalents; namely, 46% for OPC and 57% for LC. Lycopene and *β*-cryptoxanthin were associated with a lower LC risk: 50% and 59%, respectively. Lycopene, *α*-carotene, and *β*-cryptoxanthin were associated with a 26% lower risk of OCC and PC [1].

A pooled analysis of 10 case-control studies conducted in Europe, North America, and Japan [2] involved 18,207 patients (4414 with OCC and PC, 1545 with LC, and 12,248 controls). The authors noted the following cancer risk reduction associated with a total carotenoid intake: 39% for OCC and PC, and 39% for LC. The authors analyzed the impact of *β*-carotene equivalents, *β*-cryptoxanthin, lycopene, and lutein plus zeaxanthin on HNC risk. At least an 18% decrease in OCC and PC rates and a 17% decrease in the LC rate (95% CI 0–32%) was reported in patients with carotenoid consumption. The association of this protective effect in the context of smoking and drinking was also analyzed. The overall protective effect of carotenoids on HNC was significantly higher in patients with a high alcohol consumption. A significantly lower odds ratio for the combined effect of a low carotenoid intake and high alcohol consumption or smoking, compared to high carotenoid consumption and low alcohol drinking or smoking (7 (95% CI 5–9) vs. 33 (95% CI 23–49), respectively) was noted. The authors concluded that carotenoids might reduce HNC risk. The highest HNC risk was reported in smoking or drinking patients with low carotenoid consumption [2].

#### 4.1.2. Studies Investigating the Relationship between the Dietary Consumption of Carotenoids and HNC Risk

##### Prospective Cohort Studies

In a large prospective cohort study by Freedman et al. [35] involving 490,802 HNC patients, the relationship between vegetable and fruit intake and HNC risk was investigated. This study showed that the total fruit and vegetable consumption was inversely related to HNC risk. In a multivariate analysis with a model adjusted for fruit and vegetable consumption, this relationship was stronger for vegetable consumption compared to fruit intake. In addition, a decreased HNC risk was noted only in cases of whole fruit intake, but not fruit juice. In the analysis of HNC types (according to cancer locations of the oral cavity, oro-hypopharynx, or larynx), relationships between fruit and vegetables and cancer risk were similar to the overall evaluation. Regarding vegetable and fruit types, the hazard ratios (HR) for HNC risk were the following: the lowest HR was reported for rosaceae (apples, peaches, nectarines, plums, pears, and strawberries; HR = 0.60) The authors reported a similar inverse impact of fruit and vegetable intake on HNC risk in men and women.

In a large prospective cohort study by de Munter at al. [36], the association between the intake of *α*-carotene, *β*-carotene, lutein plus zeaxanthin, lycopene, and *β*-cryptoxanthin included in vegetables and fruits, and HNC risk, was analyzed. The dietary carotenoid consumption was estimated based on a food questionnaire containing the most common fruits and vegetables. These authors did not show significant associations between *α*-carotene, *β*-carotene, lycopene, and lutein plus zeaxanthin, and HNC risk.

Maasland et al. [37] in a prospective cohort study involving 120,852 patients analyzed the relationship between the intake of vegetables and fruits (including carotenoids), HNC risk, and its subtypes (OCC, OPC, and LC). The authors showed that total vegetable and fruit intake was inversely associated with overall HNC risk and its subtypes, with the strongest relationship in OCC patients. There was no significant correlation between vegetable and fruit consumption and drinking or smoking.

##### Case-Control Studies

In a case-control study by Mackerras et al. [29] involving 151 LC patients and 178 control cases, the authors showed that a low carotene intake was significantly inversely associated with LC risk (OR = 2.1). In addition, this relationship was strongest in patients with smoking cessation 2–10 years prior (OR = 5.9).

In an Italian case-control study by Franceschi et al. [30] the relationship between combined carotenoids, OCC risk, and PC risk was analyzed. The study involved 302 investigated patients and 699 control cases. The authors demonstrated that the intake of carrots, fresh tomatoes, and green peppers was related to a lower cancer risk.

Another Italian case-control study by La Vecchia et al. [31] conducted on 105 OCC and OPC patients and 1169 controls showed similar results. This study revealed a significant inverse association between intake of carrots and OCC/PC risk.

An American case-control study by Freudenheim et al. [33] involving 250 LC patients and 250 controls showed a lower LC risk in patients with a carotenoid intake. This relationship was modified by smoking. The strongest negative relationship between carotenoids and LC risk was observed in the lightest smokers. Dietary fat was most positively associated with risk among the heaviest smokers.

In the cross-sectional study by Arthur et al. [38] involving 160 HNC patients, an inverse relationship between IL-6, TNF-*α*, and IFN-γ levels and the total carotenoid intake was reported. The authors suggested that a pretreatment diet rich in vegetables, fruit, fish, poultry, and whole grains might be related to lower proinflammatory cytokine levels in HNC patients. Lower levels of circulating proinflammatory cytokines reflected lower systemic inflammation in newly diagnosed HNC patients. The circulating proinflammatory cytokines, such as (IL)-6 and tumor necrosis factor (TNF)-*α*, as mediators of the immune response, can be involved in malignant transformations, progression, and prognosis. Therefore, the lower cytokine levels may be associated with a lower probability of malignant transformation. The authors hypothesized that carotenoids mitigate the results of proinflammatory cytokines, can reduce the likelihood of metastasis, and can prolong survival.

#### 4.1.3. Studies Investigating Relationships between the Consumption and Serum Concentrations of Specific Carotenoids and HNC Risk

##### Prospective Cohort Studies

Djuric et al. [34], in their prospective cohort study, compared 60 smoking and 60 non-smoking HNC patients in order to investigate the impact of smoking on the oxidation of the carotenoid lycopene. The study showed lower serum concentrations of *α*-carotene, zeaxanthin, and 2,6-cyclolycopene-1,5-diol A (as an oxidation product of lycopene) in smoking patients compared to non-smoking patients (18%, 22%, and 8%, respectively) and similar serum concentrations of *β*-carotene and *β*-cryptoxanthin in the two compared groups. The level of lycopene was higher in smokers. In this pilot study, the authors concluded that there was no evidence that smoking status had an impact on the formation of 2,6-cyclolycopene-1,5-diol A.

In a recent prospective cohort study by Argirion et al. [39], serum micronutrients, including carotenoids in HNC patients, were analyzed. In this study, the immunohistologic expressions of CD4, CD8, CD68, CD103, CD104, and FOXP3, as markers of tumor infiltrating lymphocytes (TILs), were evaluated in tissue microarrays from 233 previously untreated HNC patients. The authors analyzed associations between the above-mentioned parameters and pretreatment dietary patterns, including carotenoids. In this study, high levels of total carotenoids (OR = 0.31; *p* < 0.0001), xanthophylls (OR = 0.12; *p* < 0.0001), and lycopene (OR = 0.36; *p* = 0.0001) were associated with significantly lower CD68+. This study showed that diet and serum carotenoids could modify TILs, as well as outcomes after diagnoses, including the overall and recurrence-free survival in HNC patients.

##### Case-Control Cohort Studies

An American case-control study by Schantz et al. [21] involving 167 HNC patients and 177 controls did not show significant associations between HNC risk and *α*-carotene consumption, *β*-carotene intake, lycopene intake (OR = 0.60; 95% CI: 0.32–1.11), and lutein intake, and showed a significant protective relationship between *β*-cryptoxanthin intake and HNC risk.

A multicenter case-control study by Negri et al. [12] that included 754 patients with OCC and OPC and 1775 control cases showed a non-significant protective influence of carotenoids on the risk of the analyzed cancers (OR = 0.61, 95% CI: 0.51–0.74; *p* = 0.91 for carotene, OR = 0.91, 95% CI: 0.79–1.03; *p* = 0.83 for lycopene). In patients with a high vitamin C consumption, carotene consumption was not associated with OCC risk, while in patients with a low consumption of vitamin C, an OR of 0.46 was reported for high carotene intake compared to those with low carotene consumption. Moreover, the relationships with vitamins C and E were stronger in patients with low carotene consumption. It should be added that in this study, the relationship of cancer risk with vitamin C and carotene consumption was independent of drinking and smoking.

In a case-control study by De Stefani et al. [17] involving 230 HNC patients and 491 control cases, the authors investigated the relationship between tomatoes, tomato products including lycopene, and cancers of the upper respiratory and digestive tracts (UADC; oral cavity, pharynx, larynx, and esophagus). This study demonstrated a strong association of lycopene with a lower HNC risk (OR = 0.22; 95% CI: 0.13–0.37). In addition, the combined impact of lycopene and the total phytosterols was related to a significantly lower HNC risk.

Bidoli et al. [18], in their case-control study involving 527 LC patients and 1297 controls, analyzed the relationship between *α*-carotene, lutein/zeaxanthin, and LC risk. The study showed a significant inverse relationship between LC risk and consumption of *α*-carotene, *β*-carotene, and lutein/zeaxanthin (OR = 0.4; 95% CI: 0.3–0.6). In addition, the cancer risk was significantly higher in smokers and drinkers with a combined low consumption of vitamin C, carotene, vitamin E, and folate (ORs were 80–170).

Another case-control study by Gallus et al. [24], involving 68 women with LC and 340 controls, showed an inverse relationship between vegetables, fruits, olive oil, and LC risk, but there was no significant relationship between carotene consumption and LC risk.

In the case-control study by Polesel et al. [40] involving 198 NPC patients and 594 controls the relationship between carotenoids in dietary intake and NPC risk was analyzed. The authors showed an inverse association between NPC risk and dietary consumption of the total carotenoids *α*-carotene and *β*-carotene.). The authors showed that carotenoids play a protective role against NPC and supported recommendations to eat fruits and vegetables to decrease the HNC risk.

In another case-control study by Bravi et al. [20] involving 768 OCC and PC patients and 2078 controls, a significantly lower cancer risk for all vegetables (OR = 0.19, for the highest vs. the lowest intake) and all fruits (OR = 0.39) was reported. Regarding carotenoids, *α*-carotene (OR = 0.51), *β*-carotene (OR = 0.28), *β*-cryptoxanthin (OR = 0.37), lutein, and zeazanthin (OR = 0.34) were related to a lower cancer risk. In addition, a combined low fruit and vegetable intake and high meat consumption, with smoking and drinking, was associated with a 10- to over 20-fold increased cancer risk.

In a case-control study by Petridou et al. [22] involving 106 OCC patients and 106 control cases, the authors analyzed the impact of diet components on OCC risk. The study did not show the significant relationship between the carotene intake and OCC risk. The mean carotene intake was similar in OCC and control patients (3610 vs. 3642 µg; *p* = 0.37).

### 4.2. Interventional Studies on the Association between Carotenoids and HNC

In a randomized, placebo-controlled, double-blinded clinical trial by Mayne et al. [41] that included 264 HNC patients, the impact of the use of *β*-carotene on the risk of second-stage HNC was analyzed. The patients underwent a curative treatment for a recent early-stage HNC. Patients were divided into two groups: the first group received 50 mg of *β*-carotene per day, and the second group received a placebo. They were observed during 90 months for the development of second primary tumors and local recurrences. The two observed groups were similar regarding the time to failure (second primary tumors plus local recurrences). There was no significant impact of *β*-carotene on second-stage HNC and mortality. Despite the lack of statistically significant results, the authors suggested a possible decrease in second-stage HNC risk in patients receiving *β*-carotene.

In another study, Mayne et al. [42] assessed the relationship between serum *α*-carotene, *β*-carotene, lycopene, lutein/zeaxanthin, total carotenoids, and mortality. The study involved 259 HNC patients. The patients received a 50 mg dose of *β*-carotene per day which was compared to a placebo. In a linear regression analysis, in models adjusted for age, serum cholesterol, time-dependent smoking, treatment regimens, the study location, and gender, only an inverse significant association between the serum concentration of lycopene and total mortality was reported. The serum concentration of *α*-carotene was inversely related to cardiovascular mortality. Lycopene, *α*-carotene, and the total carotenoids were inversely associated with mortality in non-smoking patients.

In another randomized, placebo-controlled, double-blinded clinical trial study by Bairati et al. [43], 540 HNC patients undergoing radiotherapy (RT) received *α*-tocopherol (400 IU/d) and *β*-carotene (30 mg/d) or placebos during RT and for the three years following RT. Less severe acute adverse effects during RT were reported in the supplementation group. A significant decrease in adverse effects in the combined supplementation of *α*-tocopherol and *β*-carotene was reported, regarding larynx cancer, and cancers overall in any location. There was no increase in the quality of life following this supplementation. The authors noted a higher rate of local recurrences of HNC in the supplementation group. This study demonstrated that a reduction in the severity of the treatment’s adverse effects was possible by using high doses of *α*-tocopherol and *β*-carotene during RT, but on the other hand, the use of high doses of antioxidants as an adjuvant to therapy might reduce the RT efficacy. In this study, the rate of local recurrence was 56% higher in patients receiving both *α*-tocopherol and *β*-carotene supplements compared to the placebo arm of the trial. There was even a moderate increase in the rate of local recurrence when the supplementation consisted of *α*-tocopherol alone, although it was not accompanied by a reduction in the treatment’s adverse effects.

In another study regarding this subject by Meyer et al. [44] involving 540 HNC patients undergoing RT, higher *β*-carotene dietary consumption was related to a lower rate of severe acute adverse effects. There was a tendency towards a similar effect for plasma *β*-carotene. A significantly lower rate of local recurrence was reported in patients with higher plasma *β*-carotene levels and *α*-tocopherol was not related to severe adverse effects or to cancer recurrence. The authors concluded that reduction of the risk of severe adverse RT effects and the decrease of local cancer recurrence could be achieved by high dietary *β*-carotene consumption.

The results of the above-mentioned studies by Bairati et al. [43] and Meyer et al. [44] regarding the rate of local recurrence in HNC patients are contradictory. Bairati et al. suggested that use of high doses of antioxidants as an adjuvant therapy might compromise radiotherapy efficacy, which could be related to a higher recurrence rate in this study.

### 4.3. Studies on the Relationship between Carotenoids and Survival in HNC Patients

We also found studies on the assessment of the impact of carotenoids on survival rates in HNC patients. In a case-control study by Sakhi et al. [45] involving 78 HNC patients and 100 healthy control participants, a lower serum concentration of carotenoids were reported in HNC patients compared to controls and they decreased during RT. Higher levels of total hydroperoxides (d-ROMs, a marker of oxidative stress) were noted in HNC patients compared to controls and they increased during RT. In addition, high serum concentrations of carotenoids before RT were significantly related to better progression-free survival rates (HR = 0.42; 95% CI: 0.20–0.91) in HNC patients.

In another case-control study by Sakhi et al. [46] involving 29 HNC patients and 51 healthy controls, plasma lutein, zeaxanthin, *α*-carotene, *β*-carotene, lycopene, and the total carotenoids were significantly lower in HNC patients compared to control participants. The post-RT serum concentrations of carotenoids (lutein, *α*-carotene, and *β*-carotene) were significantly positively associated with progression-free survival in HNC patients. This study showed that increasing post-RT serum carotenoid concentrations might be related to better progression-free survival rates.

## 5. Summary and Conclusions

There are not a lot of studies on the association between carotenoids and HNC in the literature. The most important above-mentioned studies are summarized in Table 1. The most important results of these studies, regarding effects of carotenoids, are presented in Figure 3. According to the published studies, carotenoids are associated with a lower HNC risk. Some studies have shown the relationship between carotenoid consumption and better survival rates in HNC patients. The role of the supplementation of carotenoids as a treatment has not been clearly described. According to some authors, it may be related to a reduction in adverse effects of RT. Reports on the reduction in the efficacy of oncological treatment by carotenoids supplementation has been also found in the literature. This problem is very important both in the prevention and management of HNC. Further prospective randomized studies regarding the impact of a controlled supplementation of carotenoids on the rates of adverse effects of RT, the efficacy of oncological treatment, and patients’ survival rates are, therefore, needed. Moreover, current studies on the role of carotenoids in HPV-related and HPV-unrelated OPC are lacking. This problem should be resolved because knowledge of the impact of carotenoids on HNC, depending on HPV status, would be helpful in the prevention and treatment of HPV-related HNC. The additional supplementation of carotenoids would be considered in the treatment protocols of these HNC subtypes. This is especially important for patients with oropharyngeal cancer.

There are some aspects that can limit the interpretation of the results of the above-mentioned studies. First of all, most of the publications concerning the relationship between carotenoids and HNC risk are case-control studies and only a few studies are randomized clinical trials. In addition, only two meta-analyses on this topic have been published so far. Moreover, the presented studies were conducted in different geographical regions and calculation methods of carotenoid intake in diets are based on different questionnaires. Databases could also be substantially different. Additionally, the methods of the assessment of the association of carotenoids and HNC risk were different, such as the overall fruit and vegetable dietary intake, the specific carotenoid dietary intake, the measurement of serum carotenoid levels, or carotenoid artificial supplements. The impact of other factors that may not have been considered, such as other nutrients, smoking, drinking, physical activity, and HPV infections should be taken into account

To our knowledge, this is the first review article with a comprehensive description of the original studies regarding the relationship between carotenoids and HNC risk. This review article summarizes the current knowledge, which can be helpful in the management of HNC patients.

Based on the presented articles, we can recommend a diet rich in vegetables and fruits for a reduction HNC risk. Further investigations are needed to answer the other questions regarding the association between carotenoids and HNC.

## Figures and Tables

**Figure 1 nutrients-14-00088-f001:**
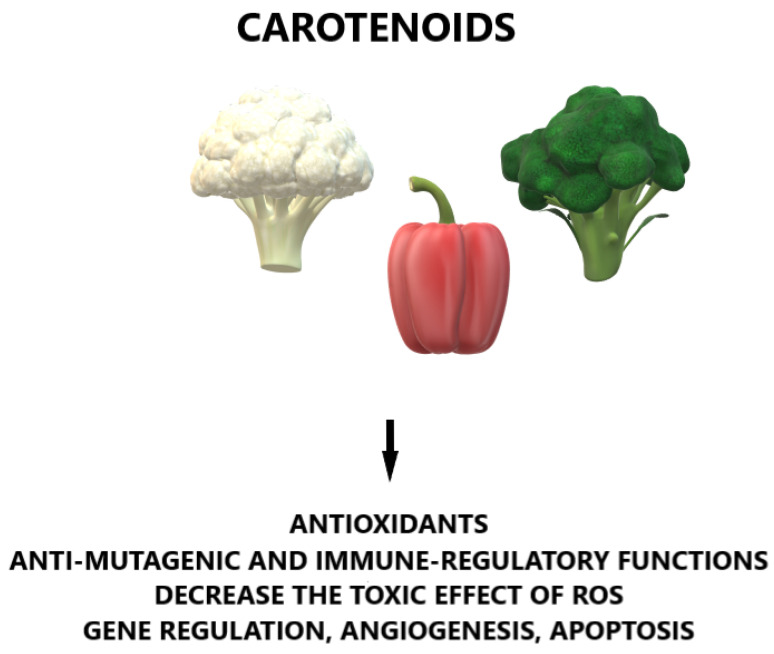
General effects of carotenoids. ROS: reactive oxygen species.

**Figure 2 nutrients-14-00088-f002:**
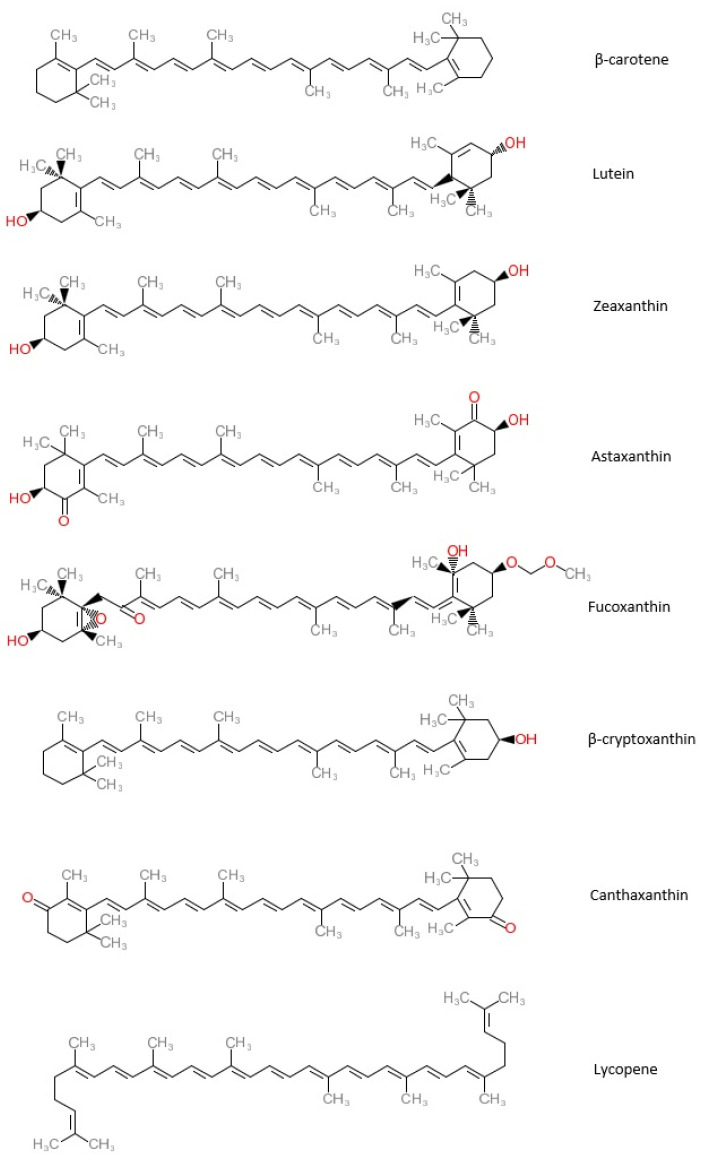
Chemical structures of the most common carotenoids.

**Figure 3 nutrients-14-00088-f003:**
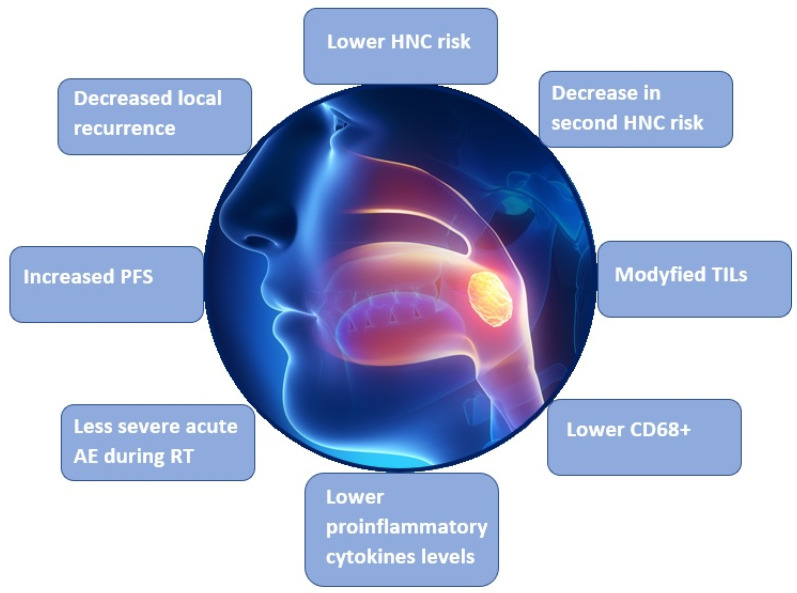
The most important results regarding effects of carotenoids in the reviewed studies. HNC: head and neck cancer; PFS: progression-free survival; TILs: tumor infiltrating lymphocytes; AE: adverse effects.

**Table 1 nutrients-14-00088-t001:** Summary of the most important studies on the relationship between carotenoids and head and neck cancer.

Authors	Year of Publication	Type of Study (Number of Patients)	Types of Carotenoids	Results/Conclusions
Systematic reviews and meta-analyses				
Leoncini et al. [1]	2015	Meta-analysis of 16 articles	Total carotenoids *α*-carotene, *β*-carotene, *β*-cryptoxanthin, lutein, zeaxanthin,	Reduction of HNC risk by carotenoids
Leoncini et al. [2]	2015	Pooled analysis of 10 articles (18,207 patients)	Total carotenoids, *α*-carotene, *β*-carotene, *β*-cryptoxanthin, lutein, zeaxanthin	Reduction of HNC risk by carotenoids
Studies investigating the relationship between dietary carotenoid consumption and HNC risk				
Prospective cohort studies				
Freedman et al. [35]	2008	Prospective cohort study (490,802 patients)	Total carotenoids	Significant inverse relationship between cancer risk and carotenoid intake
de Munter et al. [36]	2015	Prospective cohort study (450 HNC patients)	*α*-carotene, *β*-carotene, lutein, zeaxanthin, lycopene, *β*-cryptoxanthin	No significant relationships between carotenoids and HNC risk.
Maasland et al. [37]	2015	Prospective cohort study (120,852 patients)	Total carotenoids	Significant inverse relationship between total vegetable and fruit intake and overall HNC risk
Case-control studies				
Mackerras et al. [29]	1988	Case-control study (151 patients, 198 controls)	Carotene	Significant inverse relationship between carotene intake and LC risk
Franceschi et al. [30]	1991	Case-control study (302 patients, 699 controls)	Total carotenoids	Significant inverse relationship between carotenoid intake and OCC and PC risk
La Vecchia et al. [31]	1991	Case-control study (105 patients, 1169 controls)	Total carotenoids	Significant inverse relationship between carotenoid intake and OCC and OPC risk
Freudenheim et al. [33]	1991	Case-control study (250 patients, 250 controls)	Total carotenoids	Significant inverse relationship between carotenoid intake and LC risk
Arthur et al. [38]	2014	Cross-sectional study (160 patients)	Total carotenoids	Significant inverse associations between IL-6, TNF-*α*, and IFN-γ levels and quartiles of total reported carotenoid intake in HNC
Studies investigating relationships between the intake and serum concentrations of specific carotenoids and HNC risk				
Prospective cohort studies				
Djuric et al. [34]	2007	Prospective cohort study (120 patients)	Lycopene	No impact of smoking on lycopene oxidation
Argirion et al. [39]	2020	Prospective cohort study (116 patients)	Total carotenoids, xanthophylls lycopene	Significant inverse association between total carotenoids, xanthophylls, lycopene, and CD68 in HNC
Case-control studies				
Schantz et al. [21]	1997	Case-control study (167 patients, 177 controls)	*α*-carotene, *β*-carotene, *β*-cryptoxanthin, lutein, zeaxanthin	Significant protective association between *β*-cryptoxanthin intake and HNC risk
Negri et al. [12]	2000	Case-control study (754 patients, 1775 controls)	Carotene, lycopene	Significant protective impact of carotenoids on OCC and OPC risk
De Stefani et al. [17]	2000	Case-control study (230 patients, 491 controls)	Lycopene	Significant inverse relationship between lycopene and HNC risk
Bidoli et al. [18]	2003	Case-control study (230 patients, 491 controls)	*α*-carotene, *β*-carotene, lutein, zeaxanthin	Significant inverse relationship between carotenoid intake and LC risk
Gallus et al. [24]	2003	Case-control study (68 patients, 340 controls)	Carotene	Nonsignificant inverse association between carotene intake and LC risk
Polesel et al. [40]	2012	Case-control study (198 patients, 594 controls)	Total carotenoids, *α*-carotene, *β*-carotene, *β*-cryptoxanthin, lutein, zeaxanthin, lycopene	Significant inverse relationship between total carotenoids, *α*-carotene, *β*-carotene, and NPC risk
Bravi et al. [20]	2013	Case-control study (768 patients, 2078 controls)	*α*-carotene, *β*-carotene, *β*-cryptoxanthin, lutein, zeaxanthin	Significant inverse relationship between carotenoids and OC and PC risk
Interventional studies on the relationship between carotenoids and HNC				
Mayne et al. [41]	2001	Randomized, placebo-controlled, double-blinded clinical trial (264 patients)	*β*-carotene	Possible decrease in second HNC risk in supplementation
Mayne et al. [42]	2004	Randomized, placebo-controlled, double-blinded clinical trial (259 patients)	Total carotenoids *α*-carotene, *β*-carotene, lycopene, lutein, zeaxanthin	Inverse association between lycopene level and mortality, inverse association between lycopene, *α*-carotene, total carotenoids, and mortality in non-smokers
Bairati et al. [43]	2005	Randomized, placebo-controlled, double-blinded clinical trial (540 patients)	*β*-carotene	Reduction of adverse RT effects and possible reduction RT efficacy
Studies on the association between carotenoids and survival rates in HNC patients				
Prospective cohort studies				
Meyer et al. [44]	2007	Prospective cohort study (540 patients)	*β*-carotene	Reduction of adverse RT effects and better progression-free survival in patients with higher carotenoid levels
Case-control studies				
Sakhi et al. [45]	2009	Case-control study (78 HNC patients, 100 controls)	Total carotenoids, *α*-carotene, *β*-carotene, lutein, zeaxanthin, lycopene	Better progression-free survival in patients with higher carotenoid levels
Sakhi et al. [46]	2010	Case-control study (29 HNC patients, 51 controls)	Total carotenoids, *α*-carotene, *β*-carotene, lutein, zeaxanthin, lycopene	Possible better survival in patients with increasing levels of carotenoids before RT and increasing oxidative stress during RT

HNC: head and neck cancer; OCC: oral cavity cancer; PC: pharyngeal cancer; OPC: oropgaryngeal cancer; LC: laryngeal cancer; NPC: nasopharyngeal cancer; RT: radiotherapy.

## Data Availability

Not applicable.

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
