# Peer review of "The Association between Carotenoids and Head and Neck Cancer Risk"

_nutrients, 2021, doi:10.3390/nu14010088_

Round 1

Reviewer 1 Report

An article by Brewczynski et al. is generally well written. Typographical errors are still evident in the article (such as p2. Line 75; stereodescriptors R/S, E/Z should be in italics, etc.).

I would recommend the authors to add graphic illustrations to the article:

(1) Image with chemical structures of cartenoids
(2) Images highlighting the described effects of carotenoids

Author Response

Dear Reviewer,

Thank you for peer reviewing of our manuscript nutrients- 1502067, entitled " Association Between Carotenoids And Head and Neck Cancer Risk".Thank you for your questions and comments. We have fully addressed all the comments and my responses appear below. Our revised work includes corrections according to reviewers’ comments in the text. The changes, made according to reviewers’ comments, are marked up using the “Track Changes” function in the text.

We take this opportunity to express my gratitude to the reviewers for their constructive and useful remarks. Their comments allowed us to identify areas in my manuscript that needed modification.

We also thank you for allowing me to resubmit a revised copy of the manuscript.

We hope that the revised manuscript is now acceptable for publication in Nutrients.

Responses to Reviewer 1.

Comment:

An article by Brewczynski et al. is generally well written. Typographical errors are still evident in the article (such as p2. Line 75; stereodescriptors R/S, E/Z should be in italics, etc.).

Answer:

Thak you for your positive feedback. According to your suggestions the manuscript has been carefully reviewed and improved. Typographical errors have been corrected. Stereodescriptors have been written in italics.

Comment:

I would recommend the authors to add graphic illustrations to the article:

  • Image with chemical structures of cartenoids

(2) Images highlighting the described effects of carotenoids

Answer:

Thank you for your valuable suggestions. Images with chemical structures of cartenoids and highlighting the described effects of carotenoids have been added.

Reviewer 2 Report

Dietary intake of carotenoids have been associated with a reduced risk of several chronic diseases, including some types of cancer. However, intervention trials with isolated carotenoid supplements have mostly failed to confirm the postulated health benefits. Therefore, it is clear that our knowledge on carotenoid-mediated health benefits may still be incomplete.

The topic on carotenoids and head and neck cancer risk falls in the scope of this journal. This review article summarizes the current knowledge which can be helpful in the management of HNC patients, but looking at the literature and manuscripts dates, it seems that research on this topic has recently been abandoned. However, the authors have made a good collection of works on the topic, highlighting that further investigations are needed to answer the questions regarding the association between carotenoids and HNC.

Line 38:  Delete a space between “as” and “regulation”;

Lines 44,75,76,80, 130, 131:  Renumber correctly;

Line 76: Delete the point before “carotene”;

Line 81: Add the comma between “β” and “ε”;

Line 98: Delete the point before “cryptoxanthin”;

Line 101: "is" is repeated 2 times;

Line 220: Correct "questonnnaire";

Line 224: Add a space between “literature.” and “These”;

Line 286: Correct " zeazanthin ";

Line 302: Correct “fignificant”;

Line 320: Correct “supplemenation”;

Line 340: “responsed”? I believe it is "response";

Line 390: Correct “researchs”;

Table 1, page 10, last line: Correct “supplementaion”;

Table 1, page 11, last line: Correct “ncreasing”.

Furthermore, I suggest to change or delete the first two keywords because it is better to avoid repetition of words already present in the title. By using words not present in the title, you increase the chance of finding it in articles searches. 

Author Response

Dear Reviewer,

Thank you for peer reviewing of our manuscript nutrients- 1502067, entitled " Association Between Carotenoids And Head and Neck Cancer Risk".Thank you for your questions and comments. We have fully addressed all the comments and my responses appear below. Our revised work includes corrections according to reviewers’ comments in the text. The changes, made according to reviewers’ comments, are marked up using the “Track Changes” function in the text.

We take this opportunity to express my gratitude to the reviewers for their constructive and useful remarks. Their comments allowed us to identify areas in my manuscript that needed modification.

We also thank you for allowing me to resubmit a revised copy of the manuscript.

We hope that the revised manuscript is now acceptable for publication in Nutrients.

Responses to Reviewer 2.

Comment:

Dietary intake of carotenoids have been associated with a reduced risk of several chronic diseases, including some types of cancer. However, intervention trials with isolated carotenoid supplements have mostly failed to confirm the postulated health benefits. Therefore, it is clear that our knowledge on carotenoid-mediated health benefits may still be incomplete.

The topic on carotenoids and head and neck cancer risk falls in the scope of this journal. This review article summarizes the current knowledge which can be helpful in the management of HNC patients, but looking at the literature and manuscripts dates, it seems that research on this topic has recently been abandoned. However, the authors have made a good collection of works on the topic, highlighting that further investigations are needed to answer the questions regarding the association between carotenoids and HNC.

Answer:

Thak you for your positive feedback.

Comment:

Line 38:  Delete a space between “as” and “regulation”;

Answer:

Thank you for this comment. It has been deleted.

Comment:

Lines 44,75,76,80, 130, 131:  Renumber correctly;

Answer:

Thank you for this comment. In these lines, the numbers of cited publications follow the numbers of works cited in the introduction. In the line 41, references [1,7-47] have been cited. Therefore, in the paragraph 2, the numbers 48-52 are presented.

Comment:

Line 76: Delete the point before “carotene”;

Answer:

Thank you for this comment. It has been deleted.

Comment:

Line 81: Add the comma between “β” and “ε”;

Answer:

Thank you for this comment. It has been added.

Comment:

Line 98: Delete the point before “cryptoxanthin”;

Answer:

Thank you for this comment. The point “cryptoxanthin” has been left, because in this paragraph, the main types of carotenoids have been listed. The point before „lutein” has been changed (from 2.1.1. to 2.1.2) and the other carotenoids have been renumbered.

Comment:

Line 101: "is" is repeated 2 times;

Answer:

Thank you for this comment. The one word „is” has been deleted.

Comment:

Line 220: Correct "questonnnaire";

Answer:

Thank you for this comment. It has been corrected.

Comment:

Line 224: Add a space between “literature.” and “These”;

Answer:

Thank you for this comment. It has been added.

Comment:

Line 286: Correct " zeazanthin ";

Answer:

Thank you for this comment. It has been corrected.

Comment:

Line 302: Correct “fignificant”;

Answer:

Thank you for this comment. It has been corrected.

Comment:

Line 320: Correct “supplemenation”;

Answer:

Thank you for this comment. It has been corrected.

Comment:

Line 340: “responsed”? I believe it is "response";

Answer:

Thank you for this comment. It has been corrected.

Comment:

Line 390: Correct “researchs”;

Answer:

Thank you for this comment. It has been corrected.

Comment:

Table 1, page 10, last line: Correct “supplementaion”;

Answer:

Thank you for this comment. It has been corrected.

Comment:

Table 1, page 11, last line: Correct “ncreasing”.

Answer:

Thank you for this comment. It has been corrected.

Comment:

Furthermore, I suggest to change or delete the first two keywords because it is better to avoid repetition of words already present in the title. By using words not present in the title, you increase the chance of finding it in articles searches. 

Answer:

Thank you for your valuable suggestion. The first two keywords have been changed as follows:

Keywords: Oral cavity cancer; Pharyngeal cancer; Oropharyngeal cancer; Nasopharyngeal cancer; Laryngeal cancer; α-carotene; β-carotene; β-cryptoxanthin; Zeaxanthin; Lycopene; Lutein

This manuscript is a resubmission of an earlier submission. The following is a list of the peer review reports and author responses from that submission.